Unveiling the potentials of Lawsonia inermis L.: its antioxidant, antimicrobial, and anticancer potentials

Joyroy Nantikan 1
Ngiwsara Lukana Lukana@cri.or.th 2
Wannachat Siriporn 3
Mingma Ratchanee 1
Svasti Jisnuson 2
Wongchawalit Jintanart jintranart.w@ku.th 1
1 Department of Science and Bioinnovation, Faculty of Liberal Arts and Sciences, Kasetsart University , Nakhon Pathom , Thailand
2 Laboratory of Biochemistry, Chulaborn Research Institute, Chulaborn Research Institute , Bangkok , Thailand
3 Department of Animal Science, Faculty of Agricullture at Kamphaeng Saen, Kasetsart University , Kamphaeng Saen , Nakhon Pathom , Thailand
Iriti Marcello
Electronic publication date: 2025 Apr 9
Publication date: 2025
Volume: 13
Electronic Location ID: e19170
Received 2024 Nov 20; Accepted 2025 Feb 24
Copyright: ©2025 Joyroy et al.
Copyright year: 2025
Copyright holder: Joyroy et al.
License: This is an open access article distributed under the terms of the Creative Commons Attribution License, which permits unrestricted use, distribution, reproduction and adaptation in any medium and for any purpose provided that it is properly attributed. For attribution, the original author(s), title, publication source (PeerJ) and either DOI or URL of the article must be cited.
License URL: https://creativecommons.org/licenses/by/4.0/

Keywords: Lawsonia inermis, Phytochemicals, Antimicrobial activity, Antioxidant activity, Cytotoxicity, Anti-migration ability

Funding: Microbiology Division, the Department of Science and Bioinnovation, Faculty of Liberal Arts and Science, Kasetsart University Grant Years: 2024 Thailand Science Research and Innovation (TSRI) Chulabhorn Research Institute 49893/4759807 This work was supported by the Microbiology Division, the Department of Science and Bioinnovation, Faculty of Liberal Arts and Science, Kasetsart University (Grant Years: 2024); they also provided the L. inermis leaves sample. The Thailand Science Research and Innovation (TSRI), Chulabhorn Research Institute (Grant no. 49893/4759807) supported this work, the cell culture facilities experiment, and provided the use of the instrument. The funders had no role in study design, data collection and analysis, decision to publish, or preparation of the manuscript.

==============================
Background

Lawsonia inermis L., commonly known as henna, is a traditional medicinal Indian plant used for anti-dandruff and antifungal purposes. The plant is rich in phytochemicals and is believed to have significant bioactivity potential. However, limited information is available on the phytochemical compositions of L. inermis cultivars in Thailand. Therefore, this study aims to assess the phytochemical constituents and investigate the bioactivity of L. inermis extract.

Methods

L. inermis leaf extracts were prepared by macerating in ethanol (HenE), methanol (HenM), chloroform (HenC), hexane (HenH), and water boiling (HenW). The phenolic and flavonoid contents were determined by Folin-Ciocalteu and aluminum chloride colorimetric methods. High-performance liquid chromatography (HPLC) was performed to qualify polyphenolic contents. Antioxidant activities were evaluated by using 2,2-Diphenyl-1-picrylhydrazyl (DPPH), 2,2′-Azino-bis(3-ethylbenzothiazoline-6-sulfonic acid) (ABTS), and ferric reducing antioxidant power (FRAP) methods. Moreover, antibacterial activity was tested against two gram-positive and four gram-negative bacteria by the agar well diffusion and the broth dilution methods, and antifungal activity was carried out using the poisoned food technique. Additionally, the cytotoxicity of the extracts against MDA-MB-231, SW480, A549 and A549RT-eto cancer cell lines was determined by using (3-(4, 5-dimethylthiazolyl-2)-2, 5-diphenyltetrazolium bromide) (MTT) assay. The scratch wound healing assay was performed to determine the effect of anti-migration on A549 cells.

Results

Quantitative analysis revealed that HenE and HenM extracts had high phenolic and flavonoid contents. Gallic acid, catechin, ellagic acid, apigetrin, lawsone and quercetin were identified by HPLC. The HenE and HenM extracts exhibited strong antioxidant properties, and the extracts showed different inhibition growth against bacteria tested, especially B. cereus and S. aureus. In addition, all extracts had potential inhibitory activity to all fungal strains, especially HenE and Hen M, which exhibited strong antifungus activity against Penicillium sp. All extracts showed cytotoxic effects in the cell lines MDA-MB-231, SW480, A549 and A549RT-eto, except HenH. The HenE and HenM exhibited the best IC50 values of 57.33 ± 5.56 µg/ml and 65.00 ± 7.07 µg/ml against SW480 cells, respectively. The HenC, HenW, and HenH were found to suppress A549 cells migration.

Discussion and Conclusion

This study revealed that the L. inermis extracts, particularly those obtained from polar solvents (HenE and HenM), had a strong potency for antioxidant, antibacterial, and anticancer properties. Our findings highlight the valuable biological properties of extracts that can be promoted through additional investigation into their applications in Thailand for medicinal and industrial purposes.

Introduction

Currently, the world is facing an increasing incidence of chronic non-communicable diseases (NCDs) such as heart disease, stroke, diabetes, and cancer, which account for 70% of global mortality. Estimating that by 2050, half of the global population will be affected by NCDs, the urgency for effective and accessible treatment options has become more critical (WHO, 2023). Synthetic drugs are becoming less viable due to their side effects, including toxicity and the emergence of drug resistance, high costs, reliance on non-renewable resources, and adverse environmental impacts (Polianciuc et al., 2020). This situation emphasizes the requirement for alternative therapeutic approaches, especially those from medicinal plants.

The application of herbs to synthesize new drugs is gaining more attention and is considered a major source of medicine (WHO, 2023). Due to the potential in disease prevention and treatment, plant secondary metabolites have gained increasing attention across various fields, including pharmacology, medical research, food science, and agriculture (Elshafie, Camele & Mohamed, 2023). The natural compounds, particularly phytochemicals found in medicinal plants, offer a wide range of biological activities beneficial to health, for instance, antioxidant, antimicrobial, anti-inflammatory, analgesic and anticancer properties with fewer side effects compared to synthetic drugs (De Morais et al., 2020).

Lawsonia inermis L. is a flowering plant from the Lythraceae family, commonly found in tropical and subtropical regions, including Thailand. This plant is a small, densely branched shrub or tree, typically growing up to 2–6 m in height. The leaves are small, elliptical, and opposite, with a smooth, glossy surface. The flowers are fragrant, white or pinkish, and arranged in terminal panicles. The fruit is a small, brownish capsule containing numerous seeds.

L. inermis has been used in India, Sri Lanka, and the Middle East for traditional medicine and cosmetic purposes. It is rich in phytochemical compounds in various plant parts, including lawsone, flavonoids, tannins and coumarins (Khare, 2007). Their biological activities have been recorded for anti-inflammatory and antimicrobial properties (Sharma, Goel & Bhatia, 2016), antioxidants (Elansary et al., 2020) and anticancer (Singh & Luqman, 2014).

In Thailand, L. inermis has also found its place in traditional Thai medicine. The L. inermis leaf is one of the fourteen herbs used in “Ya Leuang Pid Samut” recipes for treating non-infectious diarrhea and is listed in the National List of Essential Medicines (National Essential Drug List Committee, 2011). The traditional use of henna as a natural dye appears to have nontoxic effects and seems biosecure regarding safety, antioxidant, and anti-inflammatory properties (Khantamat et al., 2021).

Medical plants in Thailand have great potential due to the high diversity of phytochemical compounds, significantly influenced by geographic factors. Therefore, this study focuses on L. inermis cultivated in Thailand and preparing crude leaf extracts using the maceration method with different organic solvents and the decoction method. Furthermore, we aim to analyze the phytochemical compositions and evaluate their various pharmacological activities, including antioxidant activity, antibacterial and antifungal effects, anti-cancer properties, and potential for inhibiting cancer cell line migration. This research may contribute to the information on the health benefits of Thai L. inermis for further development of natural therapeutic agents and reducing the utilization of synthetic drugs.

Materials & Methods

Sample collection

Aerial parts of the L. inermis were collected from Kasetsart University, Kamphaeng Saen, Nakhon Pathom Province, Thailand, at 14° 01′12.9″N and 99° 57′52.6″E, in May 2022. Plant specimens were identified by Sahanat Petchsri, PhD, staff of Botany Laboratory, Kasetsart University, Kamphaeng Saen campus. A voucher specimen (herbarium number A17819 (BCU)) was deposited in the Prof. Kasin Suvatabhandhu Herbarium (BCU) at the Department of Botany, Faculty of Science, Chulalongkorn University, Thailand. The leaves were washed several times and baked in a hot air oven at 55 °C until a constant dried weight was obtained. Then, they were finely powdered using a grinder and stored in a desiccator until used.

Preparation of L. inermis extracts

Extraction by maceration method

The powdered leaves (50 g) were soaked in 250 ml of each of four organic solvents: ethanol, methanol chloroform, and hexane. The mixture was incubated at 37 °C, 150 rpm, for 72 h, followed by filtering through Whatman filter paper No. 1. The ethanolic crude extract (HenE), methanolic crude extract (HenM), chloroform extract (HenC) and hexane extract (HenH) were concentrated by a rotary evaporator. The evaporation was performed using a Buchi R-210 rotary evaporator (Buchi R-210, Switzerland) with a flask rotation speed of 80–100 rpm and a water bath temperature of 50 °C. The vacuum pressure was set at 970 mbar using a vacuum pump to lower the boiling point of water for efficient evaporation. Subsequently, the samples were freeze-dried by lyophilizer (Telstar LyoQuest, Spain) at a condenser temperature of −70 °C and a vacuum pressure of 0.239 mbar/Pa. Dimethyl sulfoxide (DMSO) was used to dissolve the crude extracts, and all extracts were kept at 4 °C in a refrigerator before use.

Extraction by decoction method

Fifty grams of powder were boiled in 250 ml of deionized water for 15 min and cooled to room temperature. The extract was filtered, and the remaining pulp was repeatedly boiled in water for the other two rounds. All three extracts were combined and centrifuged at 8,000 rpm for 5 min, followed by filtration (Whatman No.1). Crude extract (HenW) was dried using a rotary vacuum evaporator. HenW was dissolved in distilled water for the next experiment.

Phytochemical analysis

Total phenolic content

The total phenolic content (TPC) of the extract was measured by the Folin-Ciocalteu method (Moulazadeh et al., 2021) using gallic acid as the standard. The reaction was mixed with 0.1 ml of the extracts and 0.5 ml of Folin-Ciocalteu’s reagent (Sigma-Aldrich, St. Louis, Mo, USA). The reaction was incubated in the dark at room temperature for 5 min. Then 0.4 ml of 7.5% sodium carbonate was added to the mixture and kept in the dark for 60 min. The absorbance was measured at 765 nm. The TPC was reported in milligrams of gallic acid equivalent (GAE) per gram of extract (mg GAE/g extract).

Total flavonoids content

The aluminum chloride colorimetric method was used for measuring the total flavonoids content (TFC) (Phuyal et al., 2020), with quercetin as the standard. The reaction was prepared by mixing 0.1 ml of the extract with 0.4 ml of deionized water and 0.03 ml of 5% sodium nitrite. The solution was then incubated for 5 min at room temperature in the dark, followed by adding 0.03 ml of 10% aluminum chloride and incubated for 6 min. Subsequently, 0.2 ml of 1 M sodium hydroxide and 0.24 ml of deionized water were added to the reaction. The TFC was measured by the absorbance at 510 nm and reported in milligrams of quercetin equivalent (QE) per gram of extract (mg QE/g extract).

High-performance liquid chromatography analysis

The phenolic compounds in the extracts were identified and qualified by high-performance liquid chromatography (HPLC) (modified from Francisco & Resurrection, 2009). HPLC analyses were performed using a Shimadzu HPLC Nexera system (Shimadzu LC-40 HPLC, USA) equipped with a photodiode array detector (PAD). The extracts were injected into the BDS Hypersil™ C18 column (250 mm × 4.6 mm, i.d. 5 µm) at 35 °C with a mobile phase flow rate of 1 ml/min. The mobile phase comprised 0.1% formic acid in purified water (A) and 100% acetonitrile (B). The gradient elution was performed as follows: 0–35 min, 90% A; 35–45 min, 60% A; 45–50 min, 40% A; 50–54 min, 10% A; and 54–60 min, 90% A. The phenolic compounds were detected at 280 nm and compared their retention time and UV spectral matching to standards such as gallic acid, catechin, chlorogenic acid, ellagic acid, apigetrin, lawsone and quercetin.

Antioxidant assays

DPPH radical scavenging assay

The antioxidant activity was evaluated by the 1,1-diphenyl-2-picrylphydrazyl free radical (DPPH) method (Nonthasawadsri et al., 2015). The crude extracts (7.81–1,000 µg/ml) were prepared, and each concentration, 0.3 ml, was mixed with 0.6 ml of 0.2 mM DPPH (Sigma-Aldrich, Inc., USA). The reaction was kept in the dark at room temperature for 30 min; then, the reaction mixture was measured with absorbance at 517 nm (Hitachi U-5100, Japan). Ascorbic acid was used as an antioxidant standard. The antioxidant activity was calculated as a percentage of inhibition relative to the control using the Eq. (1). The free radical scavenging properties of the extracts were also reported as a concentration required for 50% DPPH decolorization (IC50). (1) ”%DPPH scavenging activity=””Acontrol−Asample”/”Acontrol””×100”,

where; Acontrol is defined as the absorbance at 517 nm of control.

Asample is defined as the absorbance at 517 nm of the sample.

ABTS radical scavenging assay

The antioxidant activity by the 2,2′-Azino-bis(3-ethylbenzothiazoline-6-sulfonic acid) (ABTS) method was determined using the 2,2′-casino-bis (3-ethylbenzothiazoline-6-sulfuonic acid) diammonium salt (ABTS) radical cation (Re et al., 1999; Paduka et al., 2021). The solution of ABTS•+ was prepared by dissolving seven mM ABTS (Sigma-Aldrich, Inc., USA) with water, then mixed with 2.45 mM potassium persulfate (ratio of 1:0.5). The mixture was kept in the dark at room temperature for 12–16 h. Then, the ABTS•+ solution was diluted with ethanol and measured the absorbance at 734 nm until an absorbance value of 0.70 ± 0.02. The ABTS•+ solution of 1.0 ml was mixed with 0.1 ml of each extract concentration (7.81–1,000 µg/ml). The mixture was incubated at room temperature for 6 min, then the absorbance was measured at 734 nm. Trolox was used as an antioxidant standard. The antioxidant activity was calculated as a percentage of inhibition relative to the control using Eq. (2). The free radical scavenging properties were reported as a concentration required for 50% ABTS decolorization (IC50). (2) ”%ABTS scavenging activity=””Acontrol−Asample”/”Acontrol””×100”,

where; Acontrol is defined as the absorbance at 734 nm of control.

Asample is defined as the absorbance at 734 nm of the sample.

Ferric reducing antioxidant power (FRAP) assay

The ferric reducing antioxidant power (FRAP) assay was modified by Benzie & Strain (1996) and Moulazadeh et al. (2021). The FRAP reagent was prepared by mixing 300 mM acetate buffer pH 3.6 with a solution of 20 mM ferric chloride and a solution of 10 mM 2,4,6-Tris(2-pyridyl)-5-triazine (TPTZ) in 40 mM HCl (in the ratio of 10:1:1). Then, 0.05 ml of extract (1 mg/ml) was mixed with 1.5 ml of FRAP reagent solution and incubated at 37 °C for 10 min, and the absorbance was measured at 593 nm. Ferrous sulfate solution was used as standard, and the antioxidant power was reported in mmol of Fe2+/g extract.

Antibacterial activity

Microorganisms

The bacterial strains included two gram-positive bacteria, Staphylococcus aureus ATCC 27853 and Bacillus cereus TISTR 687, and four gram-negative bacteria, Escherichia coli ATCC 25922, Salmonella typhimurium ATCC 13311, Klebsiella pneumoniae and Pseudomonas aeruginosa. The bacteria were procured from the Division of Microbiology, Kasetsart University, Kamphaeng Saen Campus, Thailand.

Agar well diffusion method

The antibacterial activity of the extracts was evaluated by the agar well diffusion method (Fatahi Bafghi et al., 2022; Jitpimai et al., 2023). The bacterial strains inoculum containing 0.5 McFarland (approximately 1.5 × 108 cells/ml) were spread on Mueller Hinton Agar (MHA) plates. A volume of 0.1 ml of extract solution (300 mg/ml) was added into the wells (diameter of six mm). Ampicillin and streptomycin (10 mg/ml) were used as the positive control and DMSO as a negative control. All plates were incubated at 37 °C for 24 h. Antibacterial activity was determined and reported in millimeters (mm) of the diameter of the clear zone. Three replicates of the experiment were performed, and the data were expressed in mean ± standard deviation (SD).

Minimum inhibitory concentration (MIC) and minimum bactericidal concentration (MBC)

The minimum inhibitory concentration (MIC) of the extracts was determined using the broth dilution method described by Bussmann et al. (2010). The extracts were diluted with Mueller-Hinton Broth (MHB), which is a standardized growth medium widely used for antimicrobial susceptibility testing due to its consistent composition and ability to support the growth of non-fastidious microorganisms (Bauer et al., 1966), using a two-fold series dilution technique for the final concentration of 0.07 to 150 mg/ml. Each dilution was seeded with bacterial suspension (McFarland 0.5) and incubated by shaking at 37 °C, 150 rpm, for 24 h. The MIC was determined as the lowest concentration, where no turbidity was observed due to bacterial growth at an optical density of 600 nm. The minimum bactericidal concentration (MBC) was determined by subculturing the reaction on MHA plates and incubated at 37 °C for 18–24 h, and the lowest concentration with no visible bacterial colonies on the MHA was considered to be the MBC.

Antifungal activity

The antifungal activity was determined against Aspergillus niger, Penicillium sp., Fusarium sp. and Rhizopus sp. using a poisoned food method (Rahmoun et al., 2013). The fungal culture was inoculated on potato dextrose agar (PDA) and incubated at 25 °C for 5–7 days. All extracts were dissolved in 10% DMSO (200 mg/ml), and 0.75 ml of the extract was added to 14.25 ml of melted PDA (final concentration 10 mg/ml). The media was poured onto the plates and solidified at room temperature. PDA plates containing a final concentration of 0.5 mg/ml of Ketoconazole and 1% DMSO were used as positive and negative controls, respectively. The tip of the fungal hyphae was cut by a cork borer (diameter of six mm). The pieces of fungal strain were inoculated on a PDA and incubated at 25 °C for 4 days. Mycelial growth was recorded by measuring the diameter of the fungal colonies. The percentage of inhibition of fungal hyphae growth was calculated by the Eq. (3): (3) ”Percentage inhibition=””A−B”/”A””×100”,

where; A is the diameter of the fungus colony in the control plate.

B is the diameter of the fungus colony in the tested plates.

Cytotoxicity assay

Cell culture

Human breast adenocarcinoma cell line (MDA-MB-231), human colorectal adenocarcinoma cell line (SW480) and human lung adenocarcinoma cell line (A549) were purchased from the American Type Culture Collection (ATCC) and human lung cancer cells resistant to etoposide (A549RT-eto) was procured from Chulabhorn Research Institute. The MDA-MB-231, A549 and A549RT-eto cell lines were cultured in the Dulbecco’s Modified Eagle’s Medium (DMEM) medium. The SW480 cell lines were cultured in the Roswell Park Memorial Institute (RPMI)-1640 medium. Both media were supplemented with 10% fetal bovine serum (FBS) and 1% antibiotic-antimycotic compound. All cultures were maintained in a humidified atmosphere of 95% air and 5% CO2 at 37 °C.

Cell viability assay

Cellular viability was evaluated by (3-(4, 5-dimethylthiazolyl-2)-2, 5-diphenyltetrazolium bromide) (MTT) assay (Subhasitanont et al., 2017). The cell lines at 80% confluence were harvested and 0.1 ml inoculated in 96-well plates (5 × 103 cells/well). After 24 h, the various concentrations of the extracts (0–1,000 µg/ml) were added to the wells and incubated at 37 °C for 72 h. Each well was then replaced with a new medium containing 0.5 mg/ml of 3-(4, 5)-dimethylthaizol-2-yl-2,5-diphenyl-2H-tetrazolium bromide (MTT, Sigma-Aldrich, St. Louis, MO, USA) and incubated at 37 °C for 2 h. Finally, the media was removed from the wells, and 0.1 ml of DMSO was added to each well. The absorbance was measured at 550 nm with a microplate reader and subtracted with the absorbance at 650 nm. The number of viable cells was determined from the absorbance at 650 nm. The cell viability was plotted in a graph, and the concentration of the extract that exhibited 50% cell viability (IC50) was calculated. Data were reported as a percentage of cytotoxicity compared with the control.

Effects of the extracts on the A549 cell migration using wound healing assay

The migration capability of the A549 cells was investigated by scratch wound assay, which measures the expansion of a cell population on surfaces, as described by Grada et al. (2017) and Muniandy et al. (2018). The A549 cells (1.5 × 105 cells) were inoculated in DMEM medium in a 24-well plate and incubated at 37 °C with 5% CO2 for 24 h. Subsequently, the cells were scraped with a 0.1 ml pipette tip to create a slit in the middle of the cell layer. The scraped cell debris was washed away twice with a serum-free medium. The HenE (62.5 µg/ml), HenM (62.5 µg/ml), HenC (125 µg/ml), HenH (1,000 µg/ml), and HenW (250 µg/ml) were added at concentrations keeping cell toxicity below 20%. They were diluted with DMEM medium containing 1% antibiotic to a final volume of 0.5 ml per well. Additionally, 0.5% DMSO served as a control. Images of the scratch area were captured immediately (0 h) and post-incubation (24 h) using a microscope and analyzed with NIS Elements AR.

Data analysis

All experiments were performed in triplicate, and results were expressed as mean ± SD. One-way analysis of variance (ANOVA) followed by Duncan’s new multiple range test (DMRT) was used to analyze variance, and differences among means were compared.

Results

Extraction yield, total phenolic and total flavonoid content of the extracts

The percentage yield (%yield) and physical appearance of all crude extracts by maceration using four solvents and boiling water were shown in Table 1. Most of the extracts had a dark green color. The HenE and HenM extracts were observed as powders, while the HenC and HenH extracts presented as sticky crude extracts. The HenW extract was found to be a brown powder, with a maximum %yield of 28.82 ± 0.82%. There was no significant difference between the %yield of HenM (19.60 ± 0.46%) and HenE (18.09 ± 0.88%). HenC and HenH obtained lower %yield, at 7.32 ± 1.35 and 4.66 ± 0.84, respectively.

Table 1 Quantities of percentage yield, total phenolic content, and total flavonoid content of crude extract from L. inermis leaves.

Extract	Physical appearance	% Yield	TPC ( mg GAE/g extract )	TFC ( mg QE/g extract )	
HenE	dark green powder	18.09 ± 0.88c	163.03 ± 1.06e	464.17 ± 5.06d	
HenM	dark green powder	19.60 ± 0.46c	152.41 ± 0.79d	470.37 ± 9.86d	
HenC	dark green sticky sediment	7.32 ± 1.35b	25.13 ± 0.96b	102.58 ± 9.86c	
HenH	dark green sticky sediment	4.66 ± 0.83a	12.60 ± 0.14a	78.93 ± 3.19b	
HenW	brown powder	28.82 ± 0.82d	124.69 ± 0.70c	51.33 ± 1.15a	
Notes.

TPC, Total Phenolic Content; TFC, Total Flavonoids Content.

Different letters (a, b, c, d and e) show values that are statistically significantly different vertically at the 95% confidence level (p < 0.05); the results were expressed as mean±SD of three independent experiments, each performed in triplicate (n = 3).

The quantity results of TPC and TFC in all crude extracts were shown in Table 1. TPC of all extracts was varied and ranged from 12.60 to 163 mg GAE/g extract. The HenE extract contained the significantly highest TPC (163.03 ± 1.06 mg GAE/g extract), followed by HenM, HenW, HenC, and HenH, respectively. On the other hand, the TFC of the five extracts ranged from 51.33 to 470.37 mg QE/g. The TFC of partially L. inermis leaves depends on solvent polarity, high flavonoid content was obtained from HenM and Hen E extracts (470.37 ± 9.86 and 464.17 ± 5.06 mg QE/g, respectively) with no significant difference (p < 0.05). Subsequently, HenC, HenH, and HenW showed significant differences (p < 0.05) in TFC values as 102.58 ± 9.86, 78.93 ± 3.19, and 51.33 ± 1.15 mg QE/g, respectively.

High-performance liquid chromatography (HPLC) analysis

The phytoconstituents of the extracts were identified by HPLC analysis and compared with the retention time of seven standard substances (gallic acid, catechin, chlorogenic acid, ellagic acid, apigetrin, lawsone and quercetin) (Fig. 1). The content of the compounds present in the extracts was calculated from peak area compared to those of reference compounds. As listed in Table 2, the result revealed that ellagic acid was present in all extracts; in contrast, chlorogenic acid was absent. HenE and HenM contained six compounds and showed the highest levels of quercetin, 55.71 ± 0.11 and 56.03 ± 0.27 µg/ml, respectively. Notably, the ellagic acid, apigetrin, catechin and lawsone were detected as 32.52 ± 0.09, 4.26 ± 0.06, 7.97 ± 0.72 and 0.37 ± 0.03 µg/ml, respectively, which were the highest component in HenM extracts compared to others. While the highest gallic acid was detected in HenW (2.59 ± 0.60 µg/ml). However, the HenH extract contained two compounds, ellagic acid (21.96 ± 0.01 µg/ml) and lawsone (0.17 ± 0.01 µg/ml), with a small amount.

Figure 1 HPLC chromatograms of L. inermis leaves extracts.

(A) HenE and (B) HenM.

Table 2 Quantification of individual phenolic compounds identified in henna extract at 280 nm using HPLC system.

Extract	Quantity of phenolic compounds (µg/ml)	
	Gallic acid	Chlorogenic acid	Catechin	Ellagic acid	Apigetrin	Lawsone	Quercetin	
HenE	1.03 ± 0.02Bb	ND	7.97 ± 0.72Cd	30.79 ± 0.06De	3.08 ± 0.02Bc	0.29 ± 0.02Ba	55.71 ± 0.11Af	
HenM	1.85 ± 0.05Cb	ND	7.39 ± 0.26Cd	32.52 ± 0.09Ee	4.26 ± 0.06Cc	0.37 ± 0.03Ca	56.03 ± 0.27Bf	
HenC	0.01 ± 0.00Aa	ND	1.76 ± 0.12Bd	23.63 ± 0.01Be	0.90 ± 0.00Ac	0.22 ± 0.01Ab	ND	
HenH	ND	ND	ND	21.96 ± 0.01Ab	ND	0.17 ± 0.01Aa	ND	
HenW	2.59 ± 0.60Dc	ND	0.57 ± 0.01Aab	24.90 ± 0.66Cd	0.88 ± 0.01Ab	0.30 ± 0.08Bab	ND	
Notes.

ND, Not detected.

Uppercase letters (A–E) compare the means of the differences in the different vertical treatments. Lowercase letters (a–f) compare the mean values of the differences in the different horizontal treatments, showing a statistically significant difference at the 95% confidence level (p < 0.05); the results were expressed as mean ± SD each performed in triplicate (n = 3).

Antioxidant activity of L. inermis extracts

The results of the antioxidant activity of all extracts determined by DPPH, ABTS, and FRAP methods are summarized in Table 3. DPPH radical scavenging activity of the extracts ranged from 46.67 ± 0.58 to more than 1,000 µg/ml. The HenE and HenM extracts exhibited the best IC50 values of 46.67 µg/ml and 49.67 µg/ml, respectively, followed by HenW, HenC and HenH, as analyzed by DPPH assay, while antioxidant ascorbic acid as standard had an IC50 value of 19.76 µg/ml. The ABTS assays gave the results in terms of IC50values for reducing ABTS free radicals. HenE and HenM also showed the strongest antioxidant activity with IC50 values of 48.33 µg/ml and 47.33 µg/ml, respectively, followed by HenW, HenC and HenH, similar as determined by the DPPH assay. The reducing power of the HenE and HenM was 2.01 and 2.03 mmol Fe2+/g, respectively, which was higher than that of the HenW, HenC, and HenH extracts, respectively (Table 1).

Table 3 IC50 values for radical scavenging activities and FRAP values of the L. inermis extracts.

Extract	IC 50 of DPPH assay (µ g/ml )	IC 50 of ABTS assay (µ g/ml )	FRAP assay ( mmol of Fe 2+ /g extract )	
HenE	46.67 ± 0.58b	48.33 ± 0.58b	2.01 ± 0.12c	
HenM	49.67 ± 2.08b	47.33 ± 0.58b	2.03 ± 0.15c	
HenC	608.33 ± 2.89d	476.67 ± 7.64d	0.41 ± 0.01a	
HenH	>1,000	>1,000	0.24 ± 0.01a	
HenW	198.33 ± 2.89c	99.67 ± 2.52c	1.30 ± 0.02b	
Ascobic acid	19.76 ± 0.05a	–	–	
Trolox	–	24.34 ± 0.24a	–	
Notes.

Different letters (a, b, c, d and e) show values that are statistically significantly different vertically at the 95% confidence level (p < 0.05); the results were expressed as mean ± SD of three independent experiments, each performed in triplicate (n = 3).

Antibacterial activity of L. inermis extracts

The effectiveness of the extracts against two gram-positive bacteria (B. cereus and S. aureus) and four gram-negative bacteria (E. coli, P. aeruginosa, K. pneumonia, and S. typhimurium) was initially evaluated by the agar well diffusion method. As shown in Fig. 2 and Table 4, at a concentration of 300 mg/ml, HenE and HenM extracts exhibited anti-bacterial activity in all bacterial test stains. Those two extracts were most effective against B. cereus (18.00 ± 00 and 18.33 ± 00 mm of inhibition zone, respectively) and S. aureus (14.00 ± 67 and 17.00 ± 00 mm of inhibition zone, respectively), exceeding the efficacy of ampicillin. The HenC and HenW were found to be effective on gram-positive test strains. The anti-bacterial activity was not found in HenH for all bacterial test strains and exceptionally showed the lowest activity against B. cereus (4.00 ± 00 mm of inhibition zone).

Figure 2 Antibacterial activity of HenE extract determined by agar well diffusion.

HenE extracts with concentrations of 10 mg/mL and 300 mg/mL, DMSO, and antibiotics with concentrations of 10 mg/mL for Ampicillin and Streptomycin.

Table 4 Antibacterial activity of L. inermis extracts (300 mg/ml extract), determined by the agar well diffusion method.

Extract	Inhibition zone (mean ± standard deviation) (mm)	
	B. cereus	S. aureus	K. pneumoniae	P. aeruginosa	E. coli	S. typhimurium	
HenE	18.00 ± 0.00Ee	14.67 ± 0.58Cc	3.50 ± 0.50Aab	4.00 ± 0.00Bb	4.00 ± 0.00Bb	3.00 ± 0.00Aa	
HenM	18.33 ± 0.58Ed	17.00 ± 0.00Dc	3.50 ± 0.50Aab	4.00 ± 0.00Bb	3.00 ± 0.00Aa	3.00 ± 0.00Aa	
HenC	8.83 ± 0.29Ba	10.00 ± 0.00Ab	NI	NI	NI	NI	
HenH	4.00 ± 0.00A	NI	NI	NI	NI	NI	
HenW	12.00 ± 0.00Cb	11.50 ± 0.50Bb	NI	3.00 ± 0.00Aa	NI	NI	
Ampicillin	15.00 ± 0.00Da	36.33 ± 0.58Fe	19.33 ± 0.58Bb	NI	28.50 ± 0.50Dc	34.00 ± 0.00Cd	
Streptomycin	26.5 ± 0.50Fc	23.00 ± 0.00Ea	22.67 ± 0.58Ca	24.83 ± 0.29Cb	22.83 ± 0.29Ca	24.33 ± 0.58Bb	
Notes.

Values mean inhibition zones (mm) ± standard deviation of three replicates. Uppercase letters (A–F) compare the means of the differences in the different vertical treatments. Lowercase letters (a–e) compare the mean values of the differences in the different horizontal treatments, showing a statistically significant difference at the 95% confidence level (p < 0.05). NI, Not inhibition zone

To validate the antibacterial activity of all extracts, the minimum inhibitory concentration (MIC) and minimum bactericidal concentration (MBC) of the extracts were determined by the broth microdilution method. The MIC and MBC values in Table 5 and Fig. 3 clearly showed that all extracts, except HenH, exhibited the highest activity towards gram-positive bacteria S. aureus with MIC and MBC values of 2.34 mg/ml and 9.38 mg/ml (HenE and HenM), respectively, and 4.69 mg/ml and 37.5–75.0 mg/ml (HenC and HenW), respectively. The HenE was the most active extract with MICs and MBCs ranging from 2.34–18.8 mg/ml and 9.38–75.0 mg/ml against all bacterial strains. Among the extracts, the HenM and HenE also exhibited the most promising activity against gram-negative P. aeruginosa, K. pneumoniae, E. coli and S. typhimurium with MICs and MBCs ranging from 4.69–18.75 mg/ml and 9.38–150 mg/ml, respectively, confirming the results obtained from agar well diffusion method.

Antifungal activity of L. inermis extracts

The inhibitory effect on mycelial growth against A. niger, Penicillium sp., Fusarium sp., and Rhizopus sp. of all extracts was evaluated by poisoned food technique. The results are shown in Fig. 4; at a concentration of 10 mg/ml, all extracts suppressed the growths of all four fungal hyphae tested. The HenE and HenM extracts showed highly inhibitory activity to all fungal strains. Among those extracts, HenE exhibited pronounced antifungal activity against Penicillium sp. with almost 100% inhibition, followed by A. niger and Fusarium sp. at about 80% inhibition. The HenM extract also showed high antifungal activity towards Fusarium sp. and Penicillium sp. at percentages of 82.06 ± 0.42 and 75.67 ± 7.85, respectively, with no significant difference (p < 0.05). While the remainder of the extracts (HenC, HenH and HenW) exhibited lesser growth inhibition, between 17% and 53%.

Table 5 MIC and MBC values of L. inermis extracts, determined by broth dilution assay.

Extract	MIC (mg/ml)	MBC (mg/ml)	
	BC	SA	KP	PA	EC	ST	BC	SA	KP	PA	EC	ST	
HenE	4.69	2.34	18.75	4.69	9.38	4.69	75	9.38	75	75	37.5	37.5	
HenM	4.69	2.34	18.75	4.69	9.38	4.69	75	9.38	75	150	37.5	37.5	
HenC	9.38	4.69	37.5	9.38	18.75	9.38	150	75	>150	>150	>150	>150	
HenH	18.75	4.69	37.5	9.38	18.75	9.38	150	>150	>150	>150	>150	>150	
HenW	37.5	4.69	18.75	9.38	9.38	9.38	75	37.5	>150	150	>150	>150	
Notes.

MIC, Minimum inhibitory concentration; MBC, Minimum bactericidal concentration; BC, B. cereus; SA, S. aureus; KP, K. pneumoniae; PA, P. aeruginosa; EC, E. coli; ST, S. typhimurium.

Figure 3 Minimum bactericidal concentration of HenE extracts against bacteria.

HenE extract concentrations: 1.17, 2.34, 4.69, 9.38, 18.75, 37.5, 75, and 150 mg/mL against B. cereus, S. aureus, K. pneumoniae, P. aeruginosa, E. coli. and S. typhimurium.

Figure 4 Percent inhibition of mycelial growth by L. inermis extracts.

Cytotoxicity of L. inermis extract against cancer cell lines

The cytotoxic effects of L. inermis extracts on the MDA-MB-231, SW480, A549, and A549RT-eto cancer cell lines were evaluated using the MTT assay, as presented in Table 6. Among the tested extracts, the HenE extract demonstrated the most potent cytotoxicity across various cancer cell lines. Specifically, HenE and HenM showed greater selectivity towards the colon cancer cell line SW480, with IC50 values of 57.3 µg/ml and 65.0 µg/ml, respectively, and the lung cancer cell line A549, with IC50 values of 86.6 µg/ml and 95.0 µg/ml, respectively. These values indicated stronger cytotoxicity compared to their effects on the breast cancer cell line MDA-MB-231, which had IC50 values of 105.0 µg/ml and 160.0 µg/ml for HenE and HenM, respectively. Although the HenC and HenW extracts exhibited higher IC50 values than HenE and HenM, they also demonstrated notable selectivity towards SW480 and A549 cells. In contrast, the HenH extract showed no cytotoxicity against any of the cancer cell lines tested. Table 6 also included results on the A549RT-eto cell line, which is known for its resistance to certain chemotherapeutic agents. In this case, none of the extracts, including HenE, showed any ability to reverse drug resistance in the A549RT-eto cells.

Table 6 IC50 values of L. inermis extracts in lung cancer cells (A549), drug-resistant lung cancer cells (A549RT-eto), breast cancer (MDA-MB-231), and colon cancer (SW480) cells after 72 hours of incubation.

Extract	Inhibitory concentration at 50% (IC 50 ) (µg/ml)	
	MDA-MB-231	SW480	A549	A549RT-eto	
HenE	105.00 ± 10.80Ab	57.33 ± 5.56Aa	86.67 ± 2.49Ab	186.67 ± 17.00Ac	
HenM	160.00 ± 16.33Ab	65.00 ± 7.07Aa	95.00 ± 4.55Aa	266.67 ± 26.25Bc	
HenC	793.33 ± 77.60Cd	496.67 ± 33.99Cb	263.33 ± 12.47Ca	686.67 ± 9.43Dc	
HenH	>1,000	>1,000	>1,000	>1,000	
HenW	373.33 ± 20.55Bb	211.67 ± 11.79Ba	216.67 ± 23.57Ba	536.67 ± 17.00Cc	
Notes.

MDA-MB-231: Human breast adenocarcinoma cell line; SW480: human colorectal adenocarcinoma cell line; A549: human lung adenocarcinoma cell line; A549RT-eto: human lung cancer cells resistant to etoposide

Values are expressed as mean ± standard deviation of three replicates. Uppercase letters (A–C) indicate significant differences among vertical treatments, and lowercase letters (a–d) indicate significant differences among horizontal treatments at the 95% confidence level (p < 0.05).

Migration of A549 cancer cells induced by L. inermis extracts

The anti-migration effects of various L. inermis extracts were evaluated using a wound healing assay at non-toxic concentrations (>80% cell survival) in A549 cells. The results showed that HenC, HenW, and HenH extracts significantly inhibited A549 cell migration, with migration inhibition percentages of 28.1%, 50.8%, and 74.2%, respectively, when compared with untreated control cells. In contrast, HenE and HenM extracts did not show any inhibitory effect on cell migration (as presented in Fig. 5).

Figure 5 Migration of A549 cancer cells induced by L.inermis extract.

(A) Wound healing of the L. inermis leaf extract: A549 cells were incubated in the presence or absence of Henna extract, and images were captured at 0 and 24 h (×100 magnification). (B) The percentage migration of A549 cancer cells.

Discussion

In this study, we assessed the efficiency of different solvents and methods for extraction of phytochemicals composition and evaluation of the bioactivity of L. inermis leaves, particularly in Thai cultivar. The maceration method in different polar solvents revealed that the yield of phytochemical compounds in the leaves sample dissolved more efficiently in highly polar solvents like ethanol and methanol, than in non-polar solvents like chloroform and hexane. The yield from ethanol and methanol was not significantly different due to their closely related polarities. Our results are consistent with previous studies indicating that the maceration in ethanol yields the highest percentage compared to chloroform and water (Rahmoun et al., 2013). This was consistent with the principle that the solvents can dissolve bioactive compounds in their match polarities (Pandey & Tripathi, 2014).

According to the quantitative analysis for TPC and TFC, extracts prepared using ethanol (HenE) and methanol (HenM) contained the highest amount of phenolic and flavonoid compounds, respectively, after which the descending order was HenW, HenC and HenH. The extracts from high-polarity solvents exhibited higher TPC than those obtained through boiling water extraction, this might be due to the high temperature (75 °C) leading to the degradation of phenolic compounds (Vuong et al., 2013). TPC from HenE is higher than TPC of maceration with 70% ethanol, as stated by Moulazadeh et al. (2021). In addition, the TPC of HenM was greater than that of methanol fractionation from Kumar, Kumar & Kaur (2014). Our finding agrees with earlier research mentioned that the type of solvent and its polarity have a substantial effect on TPC extraction yield (Dirar et al., 2019). Our finding showed high flavonoid content from HenM and HenE extracts, followed by HenC, HenH, and HenW, respectively, which is consistent with the previous study, showing that the TFC of partially L. inermis leaves depends on solvent polarity, with methanol being the highest TFC value, followed by chloroform and water Kumar, Kumar & Kaur (2014). In our study, it was noticed that the TFC value derived from HenE extract was higher as compared to previous reports which used 70% extraction (197.69 ± 5.76 mg QE/g) (Moulazadeh et al., 2021). In addition, Chaves et al. (2020) reported that polar substances like flavonoids tend to dissolve better in high-polarity solvents rather than in low-polarity solvents. Generally, most flavonoids exhibit low water solubility, resulting in lower flavonoids in water extracts compared to other solvents (Chaaban et al., 2017). The solubility of TPC and TFC is influenced by the chemical composition of the plant and the polarity of the solvent utilized. As the polarity of the solvent increases, the solubility of the bioactive compound is enhanced and leading to the subsequent release of TPC (Xu & Chang, 2007). Hence, there is a necessity to improve the selection of the solvent in the procedure for extraction enhancement.

Interestingly, our study is the first report to provide an analysis of the active ingredients in L. inermis leaves grown in Thailand by HPLC, and revealed the presence of many active compounds. Following the previous findings related to TPC and TFC, the subsequent analysis concentrated on determining the specific phenolic compound present in the crude extract. Notably, the higher TFC value than TPC in our study does not imply that the absolute amount of flavonoids exceeds that of total phenols, but rather reflects the differences in the calibration standards used. The HenE and HenM extracts exhibited the highest TPC and TFC, causing the most extensive profile of phenolic compounds, which includes gallic acid, catechin, ellagic acid, apigetrin, lawsone and quercetin. These compounds are widely studied and well-known for their antioxidant, anti-inflammatory, and anticancer properties.

Several studies conducted in various countries have investigated the lawsone content in L. inermis, reporting concentrations typically ranging from 0.6% to 1.3% (w/w) (El-Shaer et al., 2007; Nounah et al., 2017; Musa et al., 2011; Hussain et al., 2011; Phirke & Saha, 2014). These variations are attributed to differences in geographic location, climatic conditions, cultivation practices, and extraction methods. For instance, Indian henna is known to exhibit higher lawsone content (1.0–1.5%) due to favorable growing conditions and traditional cultivation techniques, while henna from other regions, such as Sudan and Morocco, tends to have slightly lower concentrations (0.6–1.0%). In contrast, the lawsone content in our samples was found to be significantly lower, ranging from 0.034% to 0.074% (w/w). This discrepancy may be attributed to several factors, including environmental differences, soil composition, harvesting time, and extraction efficiency. Notably, a study conducted in Thailand reported a wide variation in lawsone content in ethanolic extracts of L. inermis leaves, ranging from 0.01% to 1.47% (w/w) across 12 different geographical regions within the country (Charoensup et al., 2017). This suggests that local environmental and agronomic factors play a critical role in determining the lawsone concentration in L. inermis. The lower lawsone content observed in our study highlights the need for further investigation into the agronomic practices, genetic variability, and extraction techniques specific to our region. Such studies could provide insights into optimizing cultivation and processing methods to enhance the lawsone yield in L. inermis. Despite the relatively low concentration of lawsone in our samples, its presence may still contribute to the observed bioactivities, as lawsone is known for biological effects such as antimicrobial and antioxidant properties (Nounah et al., 2017; Musa et al., 2011; Chaudhary, Goyal & Poonia, 2010; Batiha et al., 2024).

The antioxidant properties of the extracts were assessed by DPPH, ABTS, and FRAP; these three assays, with each employing distinct mechanism (hydrogen atom and electron transfer), were selected to assure the dependability of our findings. The highest antioxidant activity was shown by HenE and HenM, whereas HenW, HenC, and HenH had comparatively lower antioxidant activity. Both HenE and HenM, due to the ability to donate hydrogen atoms, contributed to the presence of several phenolic and flavonoid compounds for DPPH and ABTS radicals. In the FRAP assay, HenE and HenM were the strongest electron donors (reducing power) by reducing Fe (III) to Fe (II), compared to HenW, HenC, and HenH extracts, respectively. The high antioxidant properties of HenE and HenM can be attributed to the presence of high phenolic and flavonoid compounds. These compounds have aromatic rings containing hydroxyl groups, which can stabilize free radicals by transferring electrons to their structure, preventing the formation of new radicals (Pietta, 2000). For instance, quercetin, found in high levels in HenE and HenM extracts, is known for its antioxidant and anti-inflammatory effects (Murakami et al., 2015), while gallic acid is a triphenolic compound with potent antioxidant properties (Velderrain-Rodríguez et al., 2018). Therefore, our finding indicated that the antioxidant activity of extracts vary and correlates with phenolic and flavonoidcontent, as in TPC concentration results indicating the phenolic and flavonoid compounds enhance the antioxidant properties, supported by Parikh & Patel (2018).

The antibacterial activity of different L. inermis leaf extracts was assessed using the agar well diffusion method. Among these extracts, the HenE and HenM exhibit more effective bactericidal activity than other extracts against both gram-positive bacteria (B. cereus and S. aureus) and gram-negative bacteria (E. coli, P. aeruginosa, K. pneumoniae and S. typhimurium). These findings are consistent with those of Singh (2022), who demonstrated that Soxhlet extraction of L. inermis leaves using various solvents exhibited activity against S. aureus, B. subtilis, B. cereus, P. aeruginosa, and E. coli. In addition, Khan & Nasreen (2010) pointed out that methanolic extract exhibited inhibition against S. aureus, P. aeruginosa, K. pneumoniae and E. coli. This broad-spectrum activity is noteworthy, as it suggests these extracts could be valuable in treating a wide range of bacterial infections. Moreover, these findings were corroborated by the results of the MIC and MBC assays, which were performed using the broth dilution method. Our findings demonstrated that alcoholic extracts effectively inhibited a broad spectrum of pathogenic bacteria. According to HPLC analysis, these extracts showed that phytochemical compounds, such as polyphenols and flavonoids, were present in remarkably high content. Gram-positive bacteria, such as B. cereus and S. aureus, have a thick peptidoglycan layer in their cell wall, while gram-negative bacteria, like E. coli and P. aeruginosa, possess a thinner peptidoglycan layer surrounded by an outer membrane containing lipopolysaccharides. The broad-spectrum activity of henna extracts (HenE and HenM) against both types of bacteria may be attributed to the presence of polyphenols and flavonoids, which can disrupt the cell wall and membrane integrity. The free hydroxyl groups in these compounds likely interact with the peptidoglycan layer in gram-positive bacteria and the outer membrane in gram-negative bacteria, leading to inhibition of bacterial enzymes and growth. This dual mechanism underscores the potential of henna extracts as effective antibacterial agents against diverse bacterial pathogens (Mastanaiah, Prabhavathi & Varaprasad, 2011). These findings suggest that the extracts could be a viable alternative to antibiotics in treating various bacterial infections.

HenE and HenM, in particular, showed high efficacy against Penicillium sp., A. niger, and Fusarium sp., indicating that it is rich in bioactive compounds capable of targeting a wide range of fungi. This result is in agreement with previous research showing the antifungal properties of bioactive compounds, such as flavonoids, involved in the induction of reactive oxygen species (ROS) lead to fungal cell damage (Al Aboody & Mickymaray, 2020). Notably, Penicillium sp. and Fusarium sp. are resistant to various conventional antifungal agents, which underscores the potential of natural extracts, such as L. inermis extract, as alternative options for managing fungal infections (Lopez-Medina et al., 2021). The inclusion of ketoconazole as a positive control further illustrates the potential of L. inermis extract as an alternative antifungal option. Although ketoconazole (0.5 mg/ml) is a well-established drug, the effectiveness of the extracts, especially HenE and HenM at a higher concentration (10 mg/ml), suggests that natural products could complement or substitute synthetic antifungals, particularly in treating drug-resistant fungal infections. These results highlight the strong antifungal potential of L. inermis extracts, particularly HenE and HenM, which showed significant inhibitory effects across various fungal strains. Additional research is needed to pinpoint the specific bioactive compounds driving this antifungal activity and to investigate the potential synergistic effects of combining L. inermis extracts with standard antifungal medications to enhance treatment effectiveness.

The cytotoxic effects of L. inermis extracts on cancer cell lines show that certain extracts, particularly HenE and HenM, have promising effects against SW480 colon cancer cells and A549 lung cancer cells. Our study is consistent with previous research, which found that ethanolic and methanolic leaf extracts also had anticancer effects on A549 and MCF-7 breast cancer cells (Ishteyaque et al., 2020; Elansary et al., 2020). The anticancer effects of L. inermis extracts are likely due to its phenolic compounds, which are known for their antioxidant properties, especially at higher doses. In addition, the HenH extract didn’t show any cytotoxic effects, which suggests that different crude extracts have varying levels of bioactive compounds and therefore, different levels of anticancer potential. One of the major problems in cancer treatment is drug resistance. The A549RT-eto cell line, previously established by our group (Kanintronkul et al., 2011), was utilized in this study. This cell line is characterized by high expression of P-glycoprotein (P-gp), a member of the ATP-binding cassette (ABC) transporter protein superfamily, which plays a significant role in drug resistance mechanisms (Gottesman, 2002). However, the A549RT-eto cell line did not show a response to the extracts, suggesting that these extracts may not affect P–gp–mediated resistance specifically. Nevertheless, it is possible that the extracts could influence other molecular pathways involved in drug resistance; further investigation needs to be conducted to explore this potential.

This study is the first to report the inhibition of A549 cancer cell migration by L. inermis extracts. The significant reduction in migration by HenC, HenW, and HenH suggests the potential of these extracts to prevent metastasis, an essential step in cancer progression. The ability of these extracts to inhibit migration may be attributed to the presence of phenolic compounds such as ellagic acid, which was detected in high amounts in HenC and HenW, and exclusively in HenH, based on our HPLC analysis. The role of ellagic acid in inhibiting cancer cell migration is supported by previous findings. Li et al. (2016) reported that pomegranate leaf extract, which also contains ellagic acid, inhibited lung cancer cell (H1299) migration and decreased the expression of matrix metalloproteinases (MMP-2 and MMP-9). MMPs are crucial enzymes involved in cancer metastasis, as they degrade the extracellular matrix (ECM), facilitating tumor invasion and secondary tumor growth (Merchant et al., 2017). Our findings align with this mechanism, as HenC, HenW, and HenH showed inhibitory effects on migration, suggesting that ellagic acid in these extracts might contribute to the reduction of A549 cell migration by interfering with MMP activity. Since MMP-2 activation is linked to tumor development, angiogenesis, and metastatic growth, the inhibition of this enzyme could play a key role in lung cancer therapy (Togawa et al., 1999; Ke et al., 2006). Therefore, the ability of HenC, HenW, and HenH extracts to inhibit cell migration positions them as potential candidates for further exploration as safe and effective chemotherapeutic agents in the treatment of non-small cell lung carcinoma (NSCLC). These preliminary findings provide valuable insights into the potential of L. inermis leaf extracts, particularly those cultivated in Thailand, to inhibit cancer cell migration and prevent metastasis. Our findings highlight that L. inermis extracts, despite low lawsone content, exhibit significant antimicrobial and anti-migration activity. These results underscore the potential of L. inermis as a dual-purpose therapeutic agent against microbial infections and cancer metastasis.

Conclusions

The ethanol (HenE), methanol (HenM), chloroform (HenC), hexane (HenH) and water (HenW) extracts of Lawsonia inermis L. leaves showed strong potential for biological activity. The crude extracts exhibited high total phenolic and flavonoid content, testing positive for gallic acid, catechin, chlorogenic acid, ellagic acid, apigetrin, lawsone and quercetin in a preliminary phytochemical by HPLC analysis. The alcoholic extracts of HenE and HenM showed the highest antioxidant activity, followed by non-alcoholic extracts of HenW, HenC and HenH, respectively. All extracts showed different antibacterial activity profiles against the tested strains, especially HenE and HenM, which displayed the most activity towards B. cereus and S. aureus. Moreover, all extracts had potential inhibitory activity for all fungal strains, especially HenE and HenM exhibited pronounced antifungus activity against Penicillium sp. In addition, the alcoholic extracts exhibited the most potent cytotoxicity across various cancer cell lines, especially the SW480 colon cancer. The HenW, HenC and HenH extracts had the potential to inhibit cell metastasis of the A549 human lung. This finding showed new therapeutic potential for L. inermis in anti-metastatic therapy. Further study needs to focus on the mechanism involved in anti-metastasis and the major active components both in vitro and in vivo to develop a novel chemotherapeutic agent in the future. Based on the above investigations, it can be concluded that crude extracts from L. inermis leaves can be a potential source of possible herbal drug preparations for antibacterial, antifungal, and anticancer activity.

Supplemental Information

Supplemental Information 1 The raw data for quantitative of TPC and TFC, HPLC analysis, antioxidant activity, antibacterial and antifungal activity, anti-cancer activity and wound healing assay for cell migration

Special thanks are expressed to Dr. Chamrus Keawramrern for kindly supporting the bacterial test strains.

Additional Information and Declarations

Competing Interests

Author Contributions

Data Availability

The authors declare there are no competing interests.

Nantikan Joyroy performed the experiments, analyzed the data, prepared figures and/or tables, and approved the final draft.

Lukana Ngiwsara conceived and designed the experiments, analyzed the data, authored or reviewed drafts of the article, and approved the final draft.

Siriporn Wannachat performed the experiments, authored or reviewed drafts of the article, and approved the final draft.

Ratchanee Mingma analyzed the data, authored or reviewed drafts of the article, and approved the final draft.

Jisnuson Svasti analyzed the data, authored or reviewed drafts of the article, and approved the final draft.

Jintanart Wongchawalit conceived and designed the experiments, analyzed the data, authored or reviewed drafts of the article, and approved the final draft.

The following information was supplied regarding data availability:

The raw data is available in the Supplemental File.

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
