# Peer review of "Unveiling the potentials of Lawsonia inermis L.: its antioxidant, antimicrobial, and anticancer potentials"

_PeerJ, doi:10.7717/peerj.19170_

## Round 0.1 · original submission · Major Revisions

Authors should reply, in particular, to reviewer 3 on these issues:

1. species identifications
2. chemical constituents

·

Basic reporting

It Could be add more figures

Experimental design

its OKk

Validity of the findings

okk

Additional comments

its good

·

Basic reporting

Please write all abbreviations in full at the bottom of tables and charts

Experimental design

Please write a voucher number for the plant
Why did you choose the maceration and infusion method? You could do just maceration.
The dried extract could solve in DMSO easily?
In line 39, after dissolving extracts, they kept in 4ºC?
Line 115: please write the rotary properties for evaporating water.
Line 116: the extract dissolve in water easily? How was the texture of dried water extract?
Line 215: please explain MHB
Please add references for the method part.
Line 277: which Post-hoc test was used?

Validity of the findings

Please write all abbreviations in full at the bottom of tables and charts.
Don’t write any interpretation in result section
Line 398 and 399: don’t need write abbreviations in parenthesis
Table 1: why the TFC is bigger than TPC? Flavonoids are a subset of phenols and naturally their amount should be lower than that of phenols.

Additional comments

Lawson, as an important active ingredient in henna, has not received much attention in this study. Please compare it with the amount of lawson in henna from other regions of the world, and you can also write about the effects of lawson in the results of the tests.
Please also discuss the structural differences between gram-negative and gram-positive bacteria and the effect of henna extracts on them.
Line 548: “I” must be correct

Reviewer 3 ·

Basic reporting

There are several basic concerns about this MS. Here, the main ones. Henna can not be considered the name of the plant, but the product obtained by extraction. The confusion leads to the reported constituents, since lawsone is not present in the raw material, but its precursors, named hennosides, as already largely reported, but not evidently known to the authors. In fact, lawsone is not present in the graphic of the phytochemical analysis of the paper. In accordance with the international rules, the species should be identified by an expert. Who identified the plant? In absence of this identification all the research could be without any utility. In the case of this species several subspecies and cultivars are reported.

Experimental design

Se previous comments and add that the MS lacks of novelties being most of the results already present in the literature of this plant, which has been largely studied. References are clearly not adequate.

Validity of the findings

see previous comments about novelty

Additional comments

the MS is not acceptable considering the absence of several prerequisite.

---

## Round 0.2 · accepted · Accept

The authors replied satisfactorily to the most critical reviewer.

The Section Editor noted that the following minor issues should be checked: 1) L 61-62 This claim needs a supporting reference -- 2) L 112 "socked" should say "soaked" -- 3) L 406 "Thai cultivars" should say "a Thai cultivar" (only one cultivar voucher was recorded) -- 4) L 592 "my sincere" should say "our sincere"